

**A Novel Approach for Simple Statistical Analysis of High-Resolution**
**Mass Spectra**
Yanjun Zhang[1], Otso Peräkylä[1], Chao Yan[1], Liine Heikkinen[1], Mikko Äijälä[1], Kaspar R.
Daellenbach[1], Qiaozhi Zha[1], Matthieu Riva[1,2], Olga Garmash[1],  Heikki Junninen[1,3], Pentti Paatero[1],
Douglas Worsnop[1,4], and Mikael Ehn[1]
[1] Institute for Atmospheric and Earth System Research / Physics, Faculty of Science, University of
Helsinki, Helsinki, 00140, Finland
[2] Univ Lyon, Université Claude Bernard Lyon 1, CNRS, IRCELYON, F-69626, Villeurbanne,
France
[3] Institute of Physics, University of Tartu, Tartu, 50090, Estonia
[4] Aerodyne Research, Inc., Billerica, MA 01821, USA
*First author:* Yanjun Zhang & Otso Peräkylä
*Correspondence to:* Yanjun Zhang (yanjun.zhang@helsinki.fi) & Chao Yan (chao.yan@helsinki.fi)



**Abstract.** Recent advancements in atmospheric mass spectrometry provide huge amounts of new information, but at the same time present considerable challenges for the data analysts. High-resolution (HR) peak identification and separation can be effort- and time-consuming, yet still tricky and inaccurate due to the complexity of overlapping peaks, especially at larger mass-to-charge ratios. This study presents a simple and novel method, mass spectral binning combined with positive matrix factorization ('binPMF') to address these problems. Different from unit mass resolution (UMR) analysis or HR peak fitting, which represent the routine data analysis approaches for mass spectrometry datasets, binPMF divides the mass spectra into small bins and takes advantage of PMF's strength in separating different sources or processes based on different temporal patterns. In this study, we applied the novel approach to both ambient and synthetic datasets to evaluate its performance. It not only succeeded in separating overlapping ions, but was found to be sensitive to subtle variations as well. Being fast and reliable, binPMF has no requirement for a priori peak information and can save much time and effort from conventional HR peak fitting, while still utilizing nearly the full potential of HR mass spectra. In addition, we identify several future improvements and applications for binPMF, and believe it will become a powerful approach in the data analysis of mass spectra.

**Keywords.** (Atmospheric) mass spectrometry, binned positive matrix factorization (binPMF), high-resolution (HR) mass spectra, peak fitting, chemical ionization mass spectrometer (CIMS), highly oxygenated molecules (HOM)

## 1. Introduction

Volatile organic compounds (VOC) are emitted to the atmosphere both from biogenic and anthropogenic sources (Guenther et al., 1995;Wei et al., 2008). After oxidation, these gaseous species can partition to the particle phase and contribute to atmospheric organic aerosol (OA), a major component of tropospheric particulate matter (Zhang et al., 2007). The chemical components, both in particulate (OA) and gaseous phase (VOC and their oxidation products), play important roles in many atmospheric physical and chemical processes. They can deteriorate air quality causing adverse health effects, and aerosol particles can influence Earth's climate by altering the radiative balance, as well as decrease visibility (Stocker et al., 2013;Zhang et al., 2016;Pope III et al., 2009;Shiraiwa et al., 2017).



Recent instrumental advances in mass spectrometry have greatly enhanced our capability to
investigate the chemical composition and evolution of aerosol particles and their precursors. The
Aerodyne aerosol mass spectrometer (AMS) is widely applied in atmospheric research (Canagaratna
et al., 2007), measuring the bulk composition and temporal behavior of the non-refractory aerosol,
and has successfully identified different/unique OA sources utilizing factor analysis (Jimenez et al.,
2009;Zhang et al., 2011). With the development of gas-phase chemical ionization mass spectrometry
(CIMS) (Huey, 2007), and the commercially available ToF-CIMS (Bertram et al., 2011) and CI-APi-
TOF (chemical ionization atmospheric pressure interface time-of-flight mass spectrometer (Jokinen
et al., 2012)), these instruments are becoming more popular in atmospheric chemistry research. Due
to these new advances, the detection methods for aerosol precursor vapors and the understanding of
their formation mechanisms have been greatly improved. For example, the discovery of highly
oxygenated molecules (HOM) by the CI-APi-TOF has led to increased knowledge regarding
atmospheric oxidation pathways, with large implications on secondary organic aerosol (SOA) and
new particle formation (Ehn et al., 2014;Jokinen et al., 2015;Kirkby et al., 2016;Yan et al., 2016). In
particular, biogenic VOC such as monoterpenes ($C_{10}H_{16}$), promptly produce HOM upon ozonolysis,
e.g. $C_{10}H_{16}O_7$ and $C_{10}H_{16}O_9$.
While a mass spectrum can contain large amounts of information representing the highly complex
nature of the atmospheric sample, it also presents considerable challenges for the analysis and
interpretation of the data. One example of such a challenge is the identification and separation of
peaks with a similar but not identical masses. A single integer mass can contain tens of distinct ions,
with mass-to-charge ratios ($m/z$) close to each other. In all cases, specific spectral fitting techniques
are needed to resolve the overlapping peaks at the same integer mass. Typically, a least squares fit is
made      to      the      spectrum,      using      analysis      software      such      as      Squirrel/PIKA
(http://cires1.colorado.edu/jimenez-group/ToFAMSResources/ToFSoftware/), tofTools (Junninen et
al., 2010) or Tofware (https://www.tofwerk.com/software/tofware/). However, these techniques
require a pre-defined list of ions. This makes the analysis resource-intensive, and it can easily
introduce subjective bias in determining the peak list.
Figure 1 depicts a concrete example, measured by a nitrate-based CI-APi-TOF, where peak separation
is not large enough to allow unambiguous fitting of all the ions, and the final result will depend on
which ions the analyst chooses to include. As the $m/z$ increases, the number of possible ions at a
certain unit mass increases rapidly (Kroll et al., 2011;Stark et al., 2015). Too closely overlapping
peaks will sometimes lead to ambiguously fitted peaks and arbitrarily resolved ions, resulting in
unreliable separation of signals. Additionally, mass calibration errors can also affect correct peak



assignment or fitting. A few recent studies discuss in more detail the uncertainties of ion identification

and separation in HR mass spectra (Stark et al., 2015;Corbin et al., 2015;Cubison and Jimenez, 2015).

Another typical analysis approach is to utilize only the unit mass resolution, or UMR data. As opposed

to high resolution fitting, where the signals of individual ions are separated from the total measured

signal, in UMR analysis all signals at a given integer mass is integrated and treated together. This

approach is more straightforward and less subjective than HR fitting, but loses all possible high

resolution details in the spectrum (see Figure 2).

Even with perfect high resolution peak fits, a spectrum typically contains information of hundreds, if

not thousands, of ions, many of which come from similar sources. This wealth of data itself presents

a challenge for data analysis. Factor analysis enables the reduction of data dimensions and can help

to apportion the signals to factors. These factors may correspond to different sources or formation

processes. Positive matrix factorization (PMF) (Paatero and Tapper, 1994) has been widely utilized

in environmental sciences, applied to UMR and HR AMS data, succeeding in identifying multiple

OA sources (Lanz et al., 2008;Ulbrich et al., 2009;Sun et al., 2011;Zhang et al., 2011). Compared to

AMS data, PMF has been applied to CIMS data analysis much less frequently. To our knowledge,

only Yan et al. (2016) and Massoli et al. (2018) have reported PMF analysis on nitrate-based CI-APi-

TOF, utilizing UMR and HR data, respectively.

UMR-PMF cannot utilize the full information content provided by HR mass spectrometers, but is

more straightforward to apply. In contrast, accurate HR peak fitting can better preserve the

information content of the raw data than UMR, and thus provide more information to PMF, resulting

in more interpretable results. However, incorrectly fitted peaks can severely disturb the PMF

modeling and the factor interpretation. In addition, mass spectra from iodide-adduct Tof-CIMS (Lee

et al., 2014), often contain more peaks per mass than the $NO_3^-$-CI-APi-TOF, making HR fitting much

more complex (or in some cases, even unmanageable), severely limiting the potential of HR PMF.

In this study, a novel, yet simple and reliable, data analysis method, binned mass spectra combined

with PMF (binPMF), is proposed to try to tackle the abovementioned problems in both HR and UMR

PMF. Instead of using traditional UMR or HR fitting techniques to the mass spectra, we binned the

mass spectra prior to PMF analysis (Figure 2). We applied binPMF to both ambient and synthetic

datasets, succeeding in separating the key components of different sources/processes. Compared to

UMR PMF, binPMF preserves more of the high resolution information content of the mass spectra,

without the immense effort and subjectivity associated with high-resolution peak fitting. As a result,





this novel method can improve our understanding of sources/formation processes governing the
particulate and gaseous phases in more detail and in a less subjective manner.
**2. Methodology**
We divided the mass spectra into narrow bins as presented in Figure 2, and carried out PMF analysis
to extract more information from the dataset. Details on the data preparation (binning the mass
spectra), and error estimation for the PMF input are discussed in the Sections 2.2 and 2.3. To test the
performance of binPMF under different scenarios, we first constructed synthetic datasets, using a
simple one-/two-mass system (Section 2.4.1). In the second step, we applied binPMF to an ambient
dataset measured with a $NO_3^-$-CI-APi-TOF in a boreal forest site located in Southern Finland (Section

120    2.4.2).

**2.1. Positive matrix factorization**
The PMF model was developed by Paatero and Tapper (Paatero and Tapper, 1994) in the 1990s and
has been widely applied in the analysis of various types of environmental data ever since (Zhang et
al., 2017;Yan et al., 2016;Ulbrich et al., 2009;Song et al., 2007). By decomposing the observed
dataset into different factors, PMF helps to simplify the complex data matrix and extract useful
information contained within it. Compared to other common source apportionment tools, like
chemical mass balance (CMB) (Schauer et al., 1996), PMF requires no prior knowledge of source
information as essential input. Nevertheless, as a statistical method, PMF does require more data as
input, which is typically not a problem for environmental mass spectrometry datasets. The main
distinction of PMF from other factor analysis techniques is that PMF utilizes a least squares
minimization scheme weighted with data uncertainties, and non-negative constrains, to minimize the
ambiguity caused by rotation of the factors (Huang et al., 1999;Paatero and Tapper, 1994).
In PMF modelling, a measurement of chemical species is assumed to be a sum of contributions from
several relatively fixed sources/processes. The measured data matrix is broken down to two smaller
matrices, and a residual term as follows:

$$X_{(m \times n)} = TS_{(m \times p)} \times MS_{(p \times n)} + R_{(m \times n)} \tag{1}$$

where $X$ represents an $m \times n$ data matrix of original measurement for species $n$ (e.g. $m/z$) at time point
$m$, TS is the $m \times p$ time series matrix of factor contributions, MS is the $p \times n$ matrix of factor profiles
or the factor mass spectra, and $R$ is the residual between the modelled and the measured data. $p$ is the
number of factors, which needs to be determined based on the interpretability of the PMF results,





among other criteria. Thus, in PMF, the original data matrix is approximated in terms of $p$ factors,
each of which has a distinct mass spectrum and time series.
To find the solution, the PMF model utilizes uncertainty estimates for each element in the data matrix
X. These uncertainty estimates are used to weight the residuals ($R$), in order to calculate the $Q$ value
as
$$Q = \sum_{i=1}^{m} \sum_{j=1}^{n} (\frac{R_{ij}}{S_{ij}})^2 \qquad (2)$$
where $S_{ij}$ is the estimated uncertainty of species/mass $j$ at time point $i$, and $R_{ij}$ is the residual of that
mass at the same time. $Q$ is then minimized iteratively to find the mathematically optimal solution.
An expected $Q$ value ($Q_{exp}$) can be calculated as the number of non-down-weighted data values in X
minus the sum of elements in TS and MS. If the data follows the requirements of PMF, the solution
with the correct number of factors should have a $Q/Q_{exp}$ value near unity. When this is true, the
residuals on average fall within the expected uncertainties for each time point and variable. More
details about uncertainty estimation will be discussed in Section 2.3. The PMF analysis in this study
was performed with the toolkit of Source Finder (SoFi, v6.3) (Canonaco et al., 2013) by multi-linear
engine (ME-2) (Paatero, 1999). Masses with low signal-to-noise ratio (SNR < 0.2, see Section 2.3 on
error estimation) were down-weighted by a factor of 10, and masses with 0.2 < SNR < 2 were down-
weighted by a factor of 2, as suggested by Paatero and Hopke (Paatero and Hopke, 2003). The down-
weighting effect was considered in the $Q_{exp}$ calculation. In this study, PMF was operated in robust
mode, where outliers ($\left|\frac{R_{ij}}{S_{ij}}\right| > 4$) were dynamically down-weighted (Paatero, 1997).
One of the problems in any factorization analysis is rotational ambiguity, which is caused by an
infinite number of similar solutions generated by PMF (Paatero et al., 2002;Henry, 1987). Generally,
the non-negativity constraint alone is not sufficient for solution uniqueness. Rotating a certain
solution and assessing the rotated results is one possible way to determine the most physically
reasonable solution. Known source profiles or source contributions can also serve as constrains. In
addition, if there is a sufficient number of time points when the contribution of a source is nearly
zero, independent of other sources, rotational uniqueness of solutions can be achieved (Paatero et al.,
2002). The same is true if specific variables in the profiles go to zero. Otherwise, the correct solution
(correct rotation) may only be obtained by skillful use of rotational tools. Ambient measurement data
can often contain zero values in most sources/processes, greatly reducing rotational ambiguity of the
PMF results. The issue of rotational ambiguity is not explored in detail in this manuscript, as it is
common to all PMF approaches, and the main purpose here is to illustrate the new methodology of



binPMF. All the solutions shown in this study were achieved without considering their rotational
uniqueness.

## 2.2. binPMF data matrix preparation

Instead of UMR or HR fitting of the mass spectra, the mass spectra were divided into small bins after
mass calibration (Figure 2 and Figure S1 in the Supplement). Data were first linearly interpolated to
a mass interval of 0.001 Th, and then divided into bins of 0.02 Th width. At an integer mass $N$, only
the signals between $N$-0.2 Th and $N$+0.3 Th ("the signal region") were binned to avoid unnecessary
computation of masses without any signal. With the binning, there were 25 data points for each
nominal mass, instead of only one signal in UMR or several fitted peaks in HR analysis. All the
parameters mentioned above, e.g. bin width and signal region size, should be adjusted to suit the mass
spectrometer and the data being analyzed. Further details on binning procedures are discussed in
Section 3.3.

## 2.3. binPMF error matrix preparation

Beside the data matrix, an error matrix describing the expected uncertainty for each element in the
data matrix is also required as input in PMF analysis. Here, the error matrix (Polissar et al., 1998) is
estimated as

$$S_{ij} = \sigma_{ij} + \sigma_{\text{noise}} \qquad (3)$$

where the uncertainty of mass $j$ at time point $i$, $S_{ij}$, is composed of analytical uncertainty $\sigma_{ij}$ and
instrument noise $\sigma_{noise}$. $\sigma_{ij}$ is the uncertainty arising from counting statistics and is estimated as

$$\sigma_{ij} = a \times \frac{\sqrt{I}}{\sqrt{t_s}} \qquad (4)$$

in which $I$ is the signal intensity, in counts per second, $t_s$ is the averaging time in seconds and $a$ is an
empirical parameter incorporated to include unaccounted uncertainties (Allan et al., 2003;Yan et al.,
2016). In our study, we applied binPMF with CI-APi-TOF data as an example, and the same $a$ value
of 1.28 was utilized as estimated previously from laboratory experiments in the work of Yan et al.
(2016). The $\sigma_{noise}$ is calculated as the median of the standard deviation of instrument noise,
calculated from the bins between two nominal masses that should be least influenced by real signals
(the noise region), i.e. $N$+0.5 - $N$+0.8 Th (see Figure S1 in the Supplement).

## 2.4. Data sources and description





This study utilized both ambient and synthetic datasets to test the performance of binPMF. The
ambient data was collected at the SMEAR II station (Station for Measuring Ecosystem–Atmosphere
Relations (Hari and Kulmala, 2005)) in the boreal forest in Hyytiälä, Southern Finland. Located in a
rural forest area, the station has a wide range of continuous measurements of meteorology, aerosol
and gas phase properties year-round. There are no strong anthropogenic sources close to the site, but
two sawmills 5 km to southeast and the city of Tampere 60 km to the southwest. Detailed
meteorological parameters and concentrations of trace gases during this campaign have been
presented earlier (Zha et al., 2018). Before the application to ambient data, we constructed a simple
synthetic dataset, to examine how well binPMF can separate overlapping ions under different
conditions.

### 210    2.4.1. Synthetic dataset

As a first test of the performance of binPMF, we generated a series of synthetic datasets based on two
distinct sources. Each synthetic dataset $Y$ was created by summing up the signals of the two sources.
Each source consisted of a constant source profile (represented as the matrix MS), and had a unique
temporal behavior (represented as the matrix TS). Each source was the multiplication of MS (mass
spectra / source profile) and TS (time series). The two TS for the two sources were generated
randomly and independently from each other, as shown in Figure S2 in the Supplement (correlation
coefficient $R = 0.375$). To avoid rotational ambiguity (See section 2.1) in these tests, we added zero
values to the time series of the two sources, independently of each other.
As shown in Figure 3, each source profile (MS) was generated to consist of either one or two separate
peaks, covering either one or two unit masses, respectively. The peaks were generated as Gaussians
of known width and centroid position. The peaks of the different sources/profiles were partially
overlapping, with the exact overlap, i.e. the distance ($m/z$ difference) between the overlapping peaks,
being varied from one experiment to another.
Peaks in the synthetic MS profiles were first generated as perfect smooth peaks (fine $m/z$ interval of
0.00001 Th), with mass resolution of 5000 Th/Th. We define the resolution R of a peak as $R =$
$M/\Delta M$, where M is the mass of the ion, and $\Delta M$ is the full width at half maximum signal intensity,
FWHM. As an example, with R = 5000 Th/Th, an ion at $m/z$ of 300 Th will have a FWHM of 0.06
Th, corresponding to 200 ppm.  Multiplying the source profiles and the time series, we generated an
ideal data matrix. From this ideal matrix, we sampled with $m/z$ interval of 0.015 Th to simulate the
real measurement data. The interval selected was close to that typically used for the HTOF mass
analyzer on a CI-APi-TOF. After the sampling, Gaussian distributed noise, both from background



random noise and signal dependent noise were added to make up the data matrix $Y'$, point by point,
as shown in Eq. 6 below. The variance of the Gaussian distributed noise was estimated as one
hundredth of the coefficient '$c$', which is the average value of $Y$.
$$Y = \text{TS} \times \text{MS} \qquad (5)$$
$$Y'_{ij} = Y_{ij} + \text{Gaussian}\,(0, 0.01 \times c) + \text{Gaussian}\,(0, 1) \times \sqrt{Y_{ij}} \qquad (6)$$
Finally, random $m/z$ shift within ± 10 ppm was added to simulate mass calibration error, spectrum by
spectrum. This error, resulting from inaccurate conversion of the time-of-flight into mass-to-charge
ratio, is one of the main causes of ambiguous or incorrect peak assignment or fitting. In our study,
with the bin width of 0.02 Th and the mass calibration error of 10 ppm, a maximum of 15% of one
bin's signal may incorrectly shift to the adjacent bin, for a mass at 300 Th ((10 ppm × 300 Th) / 0.02
Th × 100 % = 15%). The impact of this mass shift will effectively be smaller, due to the high temporal
correlation of adjacent bins, as the signal from an ion will spread to several adjacent bins (the FWHM
is ~3 times the bin width). In the case of HR fitting of peaks, a 10 ppm mass calibration error may
cause much more dramatic changes than merely shifting 15% of the signal. There is also no reason
for ions from a given source to selectively end up at the same integer mass, meaning that the signal
is likely to be shifted to another ion from a completely different source.
Twenty-one synthetic experiments were designed, varying the mass difference between peaks ($m/z$
difference) and number of unit masses included in the MS, as shown in Table S1 and Figure 3. For
experiments 1-10, each of the two source profiles consisted of one peak (A1 and B1), both located at
the same unit mass (chosen to be 310 Th in this study), with varying separation of the peak centroids.
In experiments 11-20, we added one more peak to each profile (peaks A2 and B2), in addition to
peaks A1 and B1. The additional peaks were added at another unit mass (311 Th) and their $m/z$
difference was fixed at 0.05 Th (161 ppm), while the position of peak B1 was varied as in experiments
1-10. For experiment 21, peaks A2 and B2 were added at two different masses (311 Th and 312 Th),
corresponding to a $m/z$ difference sufficiently large that there was no meaningful overlap between
them. In the MS (i.e. mass spectra profiles), all peaks had the same intensity level initially. The
variation of the peak intensity ratio comes from variations in the time series (Figure S2 in
Supplement). The same time series for each of the two sources was used in all experiments 1-21.
With this approach of only using two masses, we purposefully provide a challenging dataset for
binPMF, as in most real datasets there would be many more masses to help to constrain the final
solutions. Nevertheless, as we will show, this simple synthetic dataset already provided a wealth of





useful information in the results attainable with binPMF, and provided a good comparison to the
traditional HR fitting approach.

### 2.4.2. Ambient dataset

The ambient dataset was measured at ground level during the Influence of Biosphere-Atmosphere
Interactions on the Reactive Nitrogen budget (IBAIRN) campaign (Zha et al, 2018) in September,
2016. The measurements were conducted using a $NO_3^-$-CI-APi-TOF that has been described in detail
elsewhere (Jokinen et al., 2012;Junninen et al., 2010;Yan et al., 2016). Here, the ambient gas-phase
molecules clustered with the nitrate ion were measured with about 4000 Th/Th mass resolving power.
Data from September $1^{st}$ to $26^{th}$, averaged to 1-hour time resolution, in the mass range of 300 - 350
Th (a typical monoterpene HOM "monomer" range, Ehn et al., 2014) were utilized for the binPMF
analysis. A baseline subtraction was applied to the mass spectra, which caused some small signals
next to large peaks to become negative. In our analysis, any $m/z$ bin where the median signal was
negative was excluded from the data matrix.

### 3.    Results and discussion

### 3.1. Synthetic dataset

### 3.1.1. Experiment settings

As introduced in Section 2.4.1, the synthetic datasets were constructed to assess the response of
binPMF to varied $m/z$ difference, peak intensity ratios, and number of masses included, as shown in
Table S1 and Figure 3. The smaller the distance between the two peaks, the harder it is to accurately
separate them with traditional HR peak identification and fitting. In our experiments, the $m/z$
difference was decreased stepwise from 0.050 Th (161 ppm) to 0.001 Th (3 ppm), in a system where
the FWHM was roughly 200 ppm.
The analysis procedure of the synthetic dataset is briefly described here. In all cases, the parameter
of interest is to see how well binPMF is able to deconvolve the adjacent peaks A1 and B1 at $m/z$ 310
Th. First, binPMF was applied to the synthetic datasets, and factors profiles (mass spectra) were
extracted. The optimal number of factors for the synthetic dataset is two, the same as the number of
sources, so only the two-factor solution was studied with binPMF. The results of the diagnostic
parameter $Q/Q_{exp}$ for each experiment are included in Table S1. Gaussian fitting was then performed
on the factor profiles to retrieve the locations of peaks A1 and B1, and thereby assess how well
binPMF was able to retrieve the original peak positions.





In addition to applying binPMF to the synthetic datasets, traditional HR peak fitting was also
conducted as comparison (by tofTools in our study). For the tofTools fits, we constrained the peak
locations and widths to those originally used for generating the data (Table S1). Peaks fitted by
tofTools and peaks fitted to the binPMF factors were compared, as well as the retrieved time series
correlation with the original datasets. More details are presented and discussed in the following
sections.
### 3.1.2. Comparison of peak fitting
We examined the performance of traditional HR fitting and binPMF by comparing their results to the
original input data. In Figure 4, the shaded areas depict the original data, the dashed lines the
traditional HR peak fitting result, and the solid lines the binPMF factors. Red and blue represent
source/factor A and B, respectively. Panels a-d (in Figure 4) show four scenarios of peak fitting results
from experiments 1, 5, 10 and 20, at the 79th time point, where the two peaks had similar signal
intensities. When the two peaks were separated by 0.05 Th (Figure 4a), both methods captured the
peak intensities quite well. However, as the $m/z$ difference narrowed, the performance of both
methods declined, with the HR fitting results deteriorating faster than those from binPMF. As $m/z$
difference reached 0.001 Th (3 ppm), the traditional HR fitting method completely failed to fit the
two peaks (panels c and d), instead attributing all the signal to just one fitted mass. In panels e-h, the
peak fitting results at the 21st time point are displayed, where the ratio of the two peaks was roughly
1:6. Here, the traditional fitting method failed to extract the two peaks already at a $m/z$ difference of
0.01 Th (30 ppm), attributing all signal to Peak B1 (panels g and h). As shown in panels d and h,
when a second set of peaks, separated by 0.05 Th, was introduced for the two sources in the datasets,
binPMF was able to utilize the temporal behavior of peaks A2 and B2, performing much better, even
in the extremely difficult cases when the $m/z$ difference for the two peaks was only 0.001 Th (3 ppm).
It is an inherent advantage of binPMF over traditional peak fitting methods that the temporal behavior
and the correlations between different variables can be utilized.
Figure 5 shows an overview of all the results of peaks fitted with binPMF. Experiments 1-10 for the
one-mass system are shown with green lines, and experiments 11-20 for the two-mass system in
yellow. Mass accuracy was calculated as the difference between fitted peak center mass and the
original mass, divided by the original mass, in ppm. When the $m/z$ difference got smaller, the mass
accuracy of peaks fitted to binPMF factors declined (Figures 5a and 5c). At a $m/z$ difference of 0.01
Th (32 ppm), the mass accuracy was -4±2 ppm and 7±2 ppm for peaks A1 and B1, respectively. The
uncertainties were estimated by repeating the analysis with 10 different random time series for the
two sources (Brown et al., 2015). For comparison, this separation approximately corresponds to that




between $C_{10}H_{16}O_7 \cdot NO_3^-$ (310.0780 Th) and $C_9H_{16}N_2O_6 \cdot NO_3^-$ (310.0892 Th). With $m/z$ difference
decreasing, the position of peak A1 (the left red peak in Figure 3), as identified by binPMF, shifted
gradually to the left, while peak B1 (the right blue peak) shifted to the right. When peaks A2 and B2
were introduced to the sources, the mass accuracy improved (< 6 ppm). The resolution of the peaks
fitted to binPMF factor profiles stayed fairly constant, but had degraded compared to the original
input data (5000 Th/Th), explained at least partially by the data binning (Figures 5b and 5d). Overall,
binPMF performs relatively well in peak separation, with reasonable mass accuracy and peak
resolution compared to the original datasets.
**3.1.3. Correlation of time series**
In addition to the peak positions, we also compared the temporal behavior of both the binPMF factors
and the time series obtained through traditional fitting, to the original time series. When $m/z$ difference
was larger than 0.02 Th (65 ppm), both methods worked similarly well in reproducing the original
time series (Figure 6). As the $m/z$ difference decreased below 0.02 Th (65 ppm), correlations
decreased rapidly (panels a and c), with that of the traditional method decreasing faster. However, as
shown by the yellow lines, when peaks A2 and B2 were added to each source profile, the time series
correlation coefficients between original data and peaks extracted by binPMF were close to unity in
experiments 11-20. The coefficients stayed similar to that from the experiment with $m/z$ difference of
0.05 Th (161 ppm), which was the fixed $m/z$ difference for the two new peaks added at 311 Th in
experiments 11-20. This means that the separation of the factor time series was mainly driven by the
additional, better separated peaks. Again, the traditional HR fitting method could not utilize the
information at 311 Th, and therefore no improvement to the peak deconvolution at 310 Th was seen.
In addition to the correlation analysis, also the assignment of absolute signal to peaks A1 and B1 was
evaluated. This was done by a linear fit (through zero) to the data points retrieved by the different
methods as a function of the original input data. The slopes of the fitted lines are plotted in Figures
6b and 6d, and show that the signal was for the most part correctly attributed to within a few percent.
The largest scatter in the determined slopes were observed for binPMF experiments with only one
mass, at low peak separations.
**3.1.4. Summary and discussion**
Based on the results shown above, binPMF was found to be as capable of separating different peaks
as traditional peak fitting techniques when the two peaks were separated by more than the mass
calibration uncertainty (yet still in all cases by less than the FWHM of the peaks). As the $m/z$
difference of the two overlapping peaks decreased, the performance of the traditional method declined


faster than that of binPMF. This was shown for signal attribution of fitted peaks and time series
correlation with original data. When masses with co-varied temporal behavior of the targeted
overlapping peaks were introduced in the dataset, the performance of binPMF improved significantly.
The peak fitting principle of the traditional method and binPMF are very different. For example,
tofTools fits peaks based on pre-determined instrument parameters (e.g. peak shape and peak width),
as well as the peak location, either as a numeric value, or a chemical composition from which the
location is calculated (Junninen et al., 2010). HR peak fitting by tofTools can be effective if the
majority of the components (peaks) are known and provided in a peak list, which is valuable
information for peak separation that was not provided to binPMF in this study. However, this
information can be hard to achieve due to unknown numbers and/or identities of all the ions at a given
mass, in combination with the limited mass resolving power of the mass spectrometer. HR peak fitting
is also sensitive to mass calibration error, increasingly so when many ions in close proximity to each
other need to be fit. On the contrary, in binPMF, peaks are separated based on the temporal variation
of masses, which is an inherent advantage of PMF, though no information of the peaks is provided
beforehand. To be more specific, a conceptual illustration is shown in Figure S3 in the Supplement.
The red peaks belong to Source A and the blue peaks to Source B. As mentioned before, the time
series of sources A and B were totally independent and random. The shaded areas (the tails of the
peaks), e.g. red shaded area in Figure S3a, contained masses that only had significant signal from
peak A1 (left red peak). Similarly, the blue shaded area in Figure S3a was mostly from peak B1. The
different temporal behaviors of the red and blue shaded areas helped the separation and correct
attribution also in the regions with overlapped signals. When the *m/z* difference of peaks A1 and B1
decreased, shown in Figure S3b, the two shaded areas also became smaller. This is the main reason
why the fitted masses of binPMF had lower mass accuracy and lower correlation coefficients
compared to the original data, as the *m/z* difference decreased.
When peaks A2 and B2 (*m/z* difference of 0.05 Th) were added in the dataset, peaks A1 and B1 were
better separated and fitted by binPMF compared to the scenarios with only one mass. This is because,
as shown in Figure S3c, the red and blue shaded areas became larger due to the addition of two more
peaks. In this case, it was peaks A2 and B2 that dominated the separation of sources A and B. In
experiment 21, three integer masses were included in the dataset. Though it was still equally difficult
for the traditional HR method to separate and fit peaks A1 and B1 with *m/z* difference of 0.001 Th (3
ppm), it was the easiest experiment for binPMF out of all the experiments because of the large *m/z*
difference of peaks A2 and B2 (1.000 Th, 3225 ppm). In experiment 21, the mass accuracies for peaks
A1 and B1 were -3.2 ppm and 2.6 ppm, respectively, and the time series correlation with original data



was 1.000 and 0.999, respectively. In most real-world applications, individual sources typically
contain multiple peaks, and the correlations of these can be utilized by binPMF.
We note once more that the results of binPMF and traditional HR peak fitting are not totally
comparable. Information about the peaks, like the exact peak centroid position, peak width
(resolution) and number of peaks, was provided to the traditional fitting method. For binPMF, no
prior information about the peaks was given, except the optimal number of factors, i.e. two.
**3.2. Ambient dataset**
With the success of binPMF for the synthetic datasets, we applied the new method to a real ambient
dataset. Here we used data collected in September 2016, from Hyytiälä in Finland. The SMEAR II
station is a forest site dominated by monoterpene ($C_{10}H_{16}$) emissions (Hakola et al., 2006). Previous
CI-APi-TOF measurements of HOM at the site (Ehn et al., 2014;Yan et al., 2016) have presented
bimodal distributions, with one mode corresponding to HOM monomers (range 300-400 Th) and the
second to HOM dimers (450-650 Th). For testing the binPMF analysis on our ambient dataset, we
selected the HOM monomer range of 300-350 Th. While the synthetic dataset primarily compared
binPMF to traditional HR fitting analysis, in this section, we compare the binPMF results with that
of traditional UMR-PMF, as employed by Yan et al. (2016). HR fitting was not performed for the
ambient dataset, for all reasons mentioned in earlier sections, including the difficulty and efforts of
producing a proper unambiguous peak list, as well the limitations of overlapping peaks.
As mentioned above, no prior knowledge was provided to PMF before the analysis. To determine the
number of factors for further analysis, we conducted runs with two to eight factors. As the number of
factors increased, more information could be extracted from the raw data. However, after the optimal
number of factors, the additional factors may split the physically reasonable factors into meaningless
fragments. There has been many studies on evaluations of PMF runs and selections of PMF factor
number (Zhang et al., 2011;Craven et al., 2012). This is an inherent challenge in any PMF analysis,
and not specific to binPMF, and therefore we do not put emphasis on this here. In this study, based
on commonly used mathematical parameters and physical interpretation, we chose the seven-factor
result, as presented below. Our main aim with this work is to present a 'proof of concept' for the
binPMF methodology, and we will therefore not provide a detailed interpretation of all the factors
(though several of the factors are easily validated based on earlier studies). The factor evolution from
two to eight factors are briefly discussed below.
From two to six factors, $Q/Q_{exp}$ showed a dramatic decrease from 6.5 to 2.7. Then for seven and eight
factors, $Q/Q_{exp}$ decreased to 2.3 and 2.0, respectively. The unexplained variation also declined from





14% to 8.8% going from two to six factors, then reached 8.0% for seven factors, and 7.6% for eight
factors. The two-factor solution first split the data into a daytime factor and a nighttime factor, with
very distinct mass spectral profiles. The daytime factor was characterized by signals at 307 Th, 311
Th, 323 Th, 339 Th and other odd masses, while the nighttime factor was dominated by 308 Th, 325
Th, 340 Th and 342 Th. The odd masses are typical signatures of day-time monoterpene-derived
organonitrates at the site, while the even masses, and specific odd masses e.g. a radical at 325 Th,
have been identified as monoterpene ozonolysis products (Ehn et al., 2014;Yan et al., 2016). As the
number of factors increased, the daytime factor was further split into new daytime factors, with
diurnal profiles having various peak times around noon or early afternoon. When the number of
factors increased to seven, a clear sawtooth shape in the diurnal trend was resolved with marker
masses at 308 Th, 324 Th, 325 Th, and 339 Th. Many of the profiles resolved in the seven-factor
solution are similar to those found by Yan et al. (2016), and separating more factors did not yield new
factors that we could interpret. Therefore, we opted to use this seven-factor result for the main
discussion below, as it provided us with enough information to evaluate the binPMF method for this
dataset.
Figure 7 shows the mass spectral profiles and factor time series for the 7-factor result, while Figure
8 displays the diurnal trends and factor contributions to the total signal. As shown in Figure 8a , the
seven factors separated by binPMF consist of one nighttime factor (Factor 1), five daytime factors
(Factors 2, 3, 4, 5 and 7) and a sawtooth-pattern factor (Factor 6).The same dataset was also analyzed
by UMR-PMF, and the corresponding seven-factor results are also included in Figures 7 and 8 for
comparison.
Overall, the results between UMR-PMF and binPMF are very similar. UMR-PMF also resolved one
clear nighttime factor, and additionally six daytime factors. For the nighttime factor, both binPMF
and UMR-PMF showed comparable temporal behavior, diurnal trend (peak at 17:00), mass spectral
profiles (peaks at 340 Th, 308 Th, 325 Th, 342 Th) and factor contribution (~ 20%). This factor has
been validated in both chamber and ambient studies to be formed from monoterpene ozonolysis (Ehn
et al., 2014;Yan et al., 2016). As shown in Figure 7a, both methods also resolved similar, though not
identical, mass spectral profiles for the other six factors, with mostly comparable time series (Figure
7b) and peak times in the diurnal trends (Figure 8a).
Despite the similarities, there also existed distinct differences between the results from binPMF and
UMR-PMF. As the most distinctive dissimilarity, binPMF Factor 6 revealed a "contamination
factor". This factor was found to be related to automated instrument zeroing every three hours, giving
rise to the distinct three-hour sawtooth pattern. The zero measurements had been removed from the





data matrix, but the zeroing system introduced some additional compounds into the sampling lines,
and the semi-volatile nature of these compounds caused them to linger, and slowly decay, in the
tubing even after the instrument had returned to sampling ambient air. binPMF accurately retrieved
the 3-hour interval of the zero measurements. However, with the same mass range (300-350 Th),
UMR-PMF failed to extract the contamination factor, regardless of the number of factors retrieved
(up to 20 factor solutions were evaluated). Instead, these contamination signals were always mixed
into the other factors. Factor 6 from UMR-PMF contributed almost twice as much as that estimated
by binPMF due to the inaccurate factor separation (Figure 8b). The time series of other factors, e.g.
Factors 5 and 7 in UMR-PMF, were clearly influenced by Factor 6. Compared to UMR-PMF, binPMF
thus showed a clear advantage in providing more information out of the data by being more sensitive
to subtle variations.
In addition to better resolving certain factors from the data, the binPMF mass spectral profiles will
still contain more information than visible in Figure 7, due to the multiple bins at each unit mass. As
an example, binPMF Factor 6 showed masses with clear negative mass defects, e.g. at 324 Th and
339 Th (Figure 9). We identified many ions in this factor as different fluorinated carboxylic acids,
common interference signals in negative ion CIMS, outgassing from e.g. Teflon tubing (Brown et al.,
2015;Ehn et al., 2012;Heinritzi et al., 2016). The exact source of these products in our setup was not
established, but it is not surprising that the additional valves, filters and/or tubing in the zeroing line
could have caused this type of signal to be introduced to the instrument with the zero air. In general,
this finding highlights the usefulness of the binPMF approach, where factor separation can be
performed first, and the specific factor profiles can be utilized in interpreting the physical meaning
of the different factors. This is in complete contrast to the more traditional approach, where all ions
need to be identified first, and only then can HR PMF be attempted. As not all ions are going to be
observable at all times, many ions may remain unidentified. For example, if peak identification would
only have been done during periods when the HOM signals were high, as in the case shown in Figure
9a, the fluorinated ion at 339 Th would not have been found (contributing only 0.45% to the total
signal at this time point), even though it on average contributes nearly 10% of the signal at this mass
over the entire campaign. binPMF, on the other hand, utilized the full dataset for the identification,
and was able to separate several ions at 339 Th. By fitting gaussian signals to the factor profiles,
similar to the synthetic data in section 3.1.2, we see that the two major peaks were fitted with decent
resolution (Figure 9). Also the contamination factor (Factor 6) was clearly separated and fitted, and
the resolution (3136 Th/Th) is slightly underestimated by the fit, as only one gaussian was fitted to
each profile, yet there are clearly more than one ion at 339 Th in Factor 6. As shown in Figure 9c,



there is also clear indication that Factors 3 and 5, which together make up as much signal at 339 Th
as the contamination Factor 6, mainly contain signals from another molecule ($C_{10}H_{13}O_9$) than the
dominant signals at this mass ($C_{10}H_{15}NO_8$). However, further work will be needed to validate this.
Factor 1 has marginal contribution to the signal at 339 Th (as seen in Figure 9b) throughout the
campaign, and we expect it does not contain useful signal, as is suggested by the unreasonably high
resolution, i.e. narrow peak width, of the fitted peak. The resolving power of the instrument was
around 4000 Th/Th, and thus any apparent peak resolution above that will be unrealistic. However,
as this factor contains signal at the outer edges of the main peaks at this *m/z*, it is possible that this
factor relates to some instrumental variability affecting the peak shape. This is highly speculative, but
such a phenomenon may be worth looking into in later studies utilizing binPMF. In summary,
resolving multi-overlapping peaks by traditional methods is time-consuming and can be tricky and
ambigous. Here, binPMF greatly simplified this problem, by providing additional separation between
the ions.

### 3.3. Future improvements and applications

The new technique for mass spectra analysis, binPMF, as presented above, shows clear promise in
utilizing HR information while saving time and effort, as well as decreasing ambiguity related to
conventional HR peak fitting. It is also more sensitive to subtle variation than standard UMR analysis.
We consider this study a succesful proof-of-concept, and note that several future improvements and
applications are still foreseeable. We list some of these below:
(1) **Varied bin width.** The full width at half maximum of an individual peak in a mass

spectrometer is mass dependent, with peaks getting wider at higher masses. In binning the

mass spectrum with a constant bin width, like in this study, the average number of bins per

peak increases as a function of mass. To represent the peaks in a comparable manner, the bin

width should thus be dependent on the mass. Varying the bin width as a function of the mass,

and the mass resolution of the instrument, would enable a constant number of bins (e.g., seven)

514        per peak. Too few bins per peak would mean that we may lose valuable information in the

binning, while too many points per peak would lead to an unnecessarily high number of

variables, without noticeable gain in information content. This would also result in high

computational cost. If targeting 7 bins per peak, then the function for determining bin width

based on *m/z* and resolution (R, which is mass-dependent) could be

$$\because \ R \ (m/z) = \frac{m/z}{\Delta m}$$
$$Bin \ width \ \times 7 = 2 \times \Delta m$$



$$\therefore \ Bin\ width = \frac{2}{7} \times \frac{m/z}{R(m/z)}$$

$\Delta m$ is the full peak width at half maximum signal intensity. If we consider an instrument with
approximate constant resolution of 5000 Th/Th for masses from 200 Th to 600 Th, the bin
width at 200 Th and 600 Th should be around 0.01 Th and 0.03 Th, respectively.
(2) **Optimization of binning region**. Similarly to bin width, the binning region, i.e. the signal
region ([N-0.2,N+0.3] in this study, introduced in section 2.2), should also be mass-dependent.
Due to the widening of the peaks with increasing mass, the binning region should also get
wider. In addition, the typical mass defect of measured ions typically varies with mass. This
means that the binning regions should not necessarily be defined with respect to the integer
masses, but to some chosen mass defect. Another approach would be to bin all the data, and
remove the bins not meeting a certain criterion, such as one related to the signal to noise ratio
in that bin, afterwards. In this case, there would be no need for a pre-defined mass defect or
region width, and one could utilize the signals that do not fall within the expected regions.
(3) **Error estimation.** Good error estimation is crucial to PMF calculation. In addition to the two
error estimation terms discussed in section 2.3, $\sigma_{ij}$ and $\sigma_{noise}$, a third form of error, caused by
mass calibration variation could also be considered for error estimation.
(4) **Multi-peak fitting.** As discussed, peak identification is one of the most time-consuming and
potentially ambiguous tasks in HR analysis, and with binPMF this may not always be a
necessary task. However, as binPMF often resolves several peaks (chemical components) at
each integer mass, peak identification can be made easier if peak identification is constrained
to several binPMF factor profiles rather than just the initial HR spectrum. The optimal
approach for this will be the target of a future study.
Most likely several other improvements to the approach will be identified in future studies, and
simplicity of the analysis remains a critical consideration. We propose that binPMF is a good tool for
initial exploration of new datasets, at which stage optimizing all parameters is not necessarily crucial,
if the results can help guide further analysis directions. However, for maximizing the information
content that can be extracted from a given data set, optimized routines are important.
**4. Conclusions**
While recent advances in mass spectrometry have greatly enhanced our understanding of atmospheric
chemistry, the increased information content in mass spectra also brings difficulties and challenges
to the data analysis. Peak identification and separation can be challenging and ambiguous, as well as
extremely time-consuming and involving large uncertainties. Constructing peak-lists, i.e. deciding



which ions to fit to the mass spectra, and validating the results is becoming one of the most labor-intensive parts of the entire work. In this study, we propose a simple and reliable method, binPMF, to try to avoid many of these problems, while still being able to distinguish different chemical pathways/sources in the atmosphere.

Different from traditional analysis, binned positive matrix factorization (binPMF), divides the mass spectra into smaller bins, before applying PMF to distinguish different types of factors and behavior in the data. This method utilizes more available information than classical UMR-PMF, and requires no prior peak information as in the case of traditional HR-PMF. We applied binPMF successfully to both ambient and synthetic datasets to test its usefulness under different circumstances.

Traditional HR analysis fits peaks to each mass according to a pre-defined list, and is not able to utilize any information across masses or time. In our analysis of a simple synthetic data set with two overlapping ions at a single integer mass, we found that binPMF was able to separate the contributions of each ion even in cases where the HR analysis failed completely. This was the case for overlapping ions where binPMF had help in constraining the time series from another integer mass. When applied to an ambient dataset of HOM measured by a CI-APi-TOF, binPMF identified more physically meaningful factors than UMR-PMF. Additionally, for factors where the two PMF approaches agreed, binPMF still contained more mass spectral information for ion identification, as compared to UMR-PMF.

We provide a proof-of-concept for the utility of binPMF, showing that it can outperform the two traditional analysis approaches, UMR and HR. We identify several future improvements and applications for binPMF, including an approach to greatly facilitate the time-consuming process of peak-list construction. We expect binPMF to become a powerful tool in the data exploration and analysis of mass spectra.

**Acknowledgements**

This research was supported by the European Research Council (Grant 638703-COALA), the Academy of Finland (grants 317380 and 320094), and the Vilho, Yrjö and Kalle Väisälä Foundation. K.R.D. acknowledges support by the Swiss National Science postdoc mobility grant P2EZP2_181599. We thank the tofTools team for providing tools for mass spectrometry data analysis. The personnel of the Hyytiälä forestry field station are acknowledged for help during field measurements.





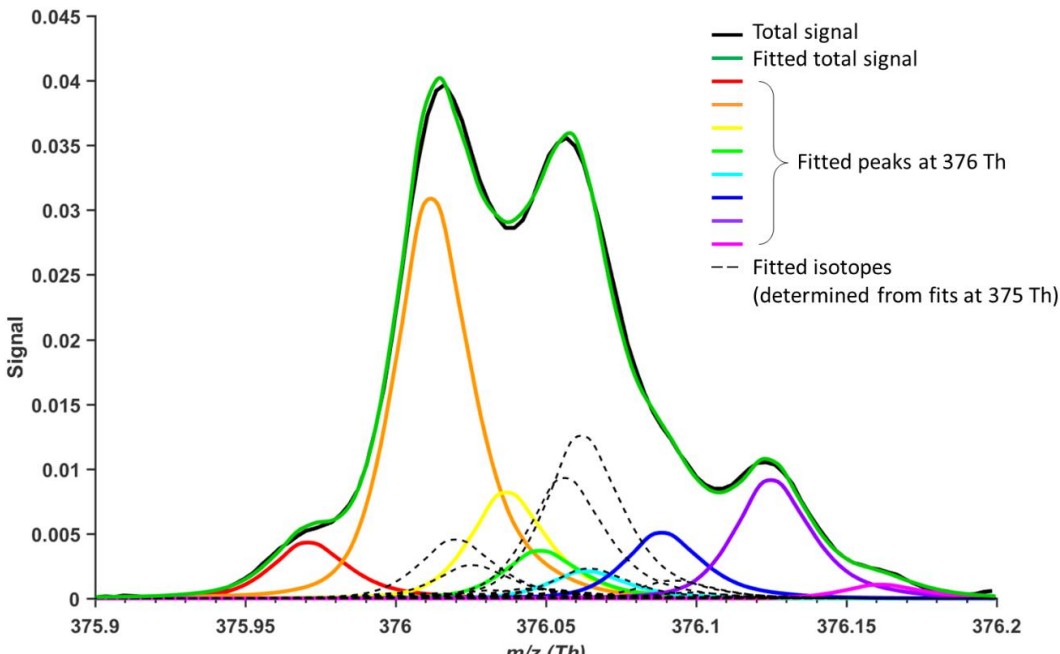

584

Figure 1. Example of traditional HR peak fitting. Potential peak fitting at *m/z* 376 Th (10-h average) in an atmospheric simulation chamber during a monoterpene ozonolysis experiment, utilizing a nitrate-based CI-APi-TOF (resolving power of 13000 Th/Th). Even a minor shift in the mass axis calibration could cause the signals of especially the yellow, green and blue peaks to change dramatically. Similarly, adding or removing an ion would alter the amount of signal attributed to the other fitted peaks.




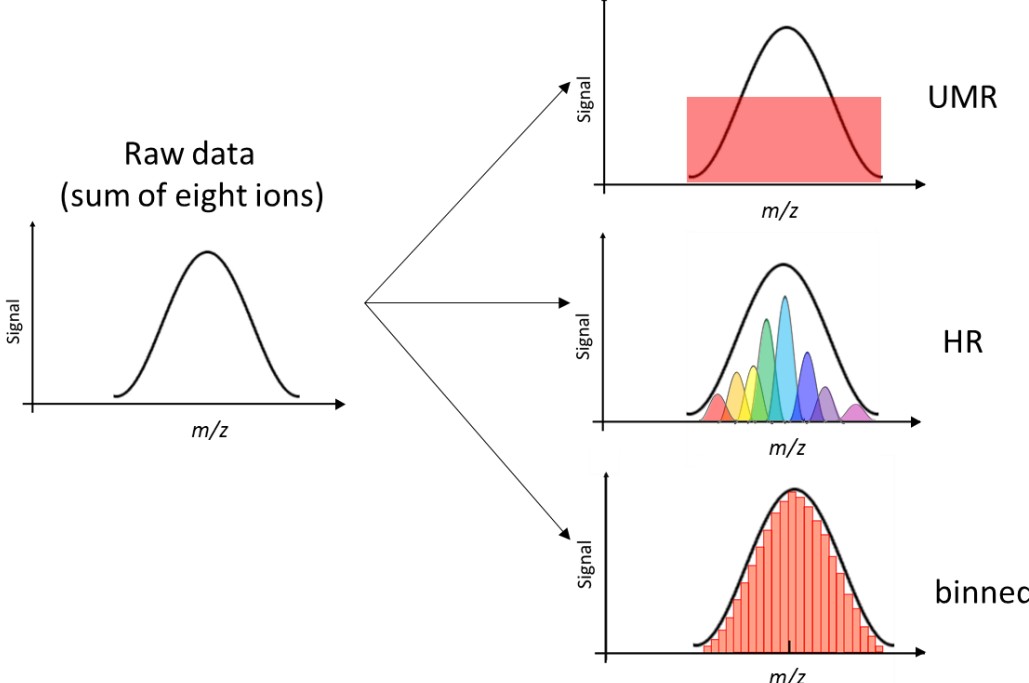

591

Figure 2. Conceptual comparison of traditional methods (UMR and HR) and binned mass spectra for

PMF analysis. The raw data signal is shown in the left and contains eight ions. By UMR analysis, the

information of the eight ions is totally lost. Using an analyst-determined peak-list, HR analysis

attempts to separate signals at this mass by fitting selected ions. By binning the spectra, we utilize the

HR information without any a priori information required.





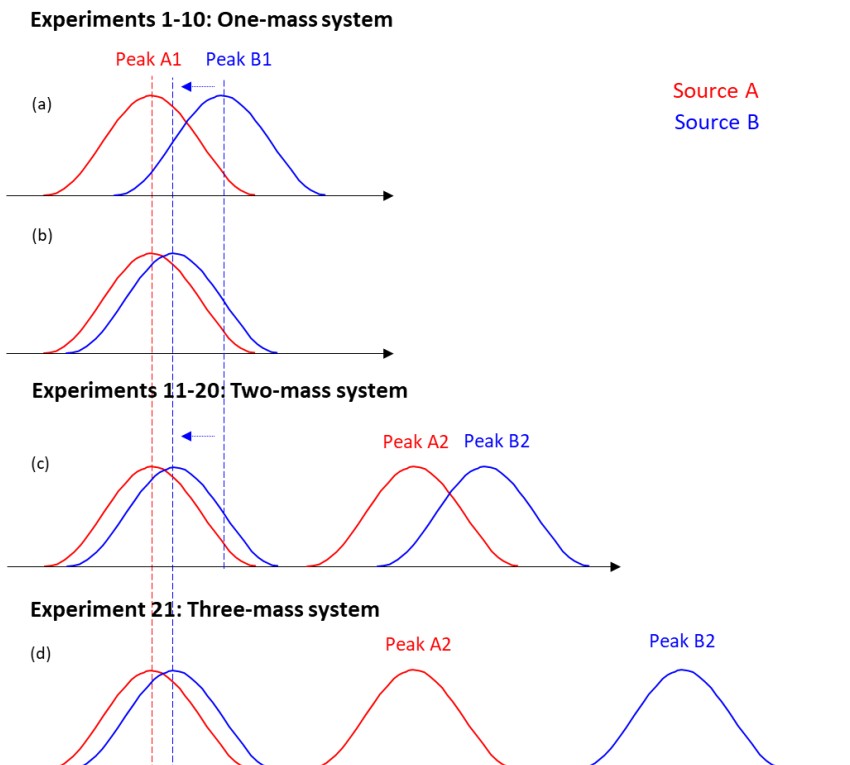

597

Figure 3. Conceptual schematic diagram for the synthetic datasets. Panels a and b describe experiments 1-10 in the one-mass system, panel c is experiments 11-20 in the two-mass system. Panel d shows experiment 21, with peaks A2 and B2 at separate integer masses (see text for details).





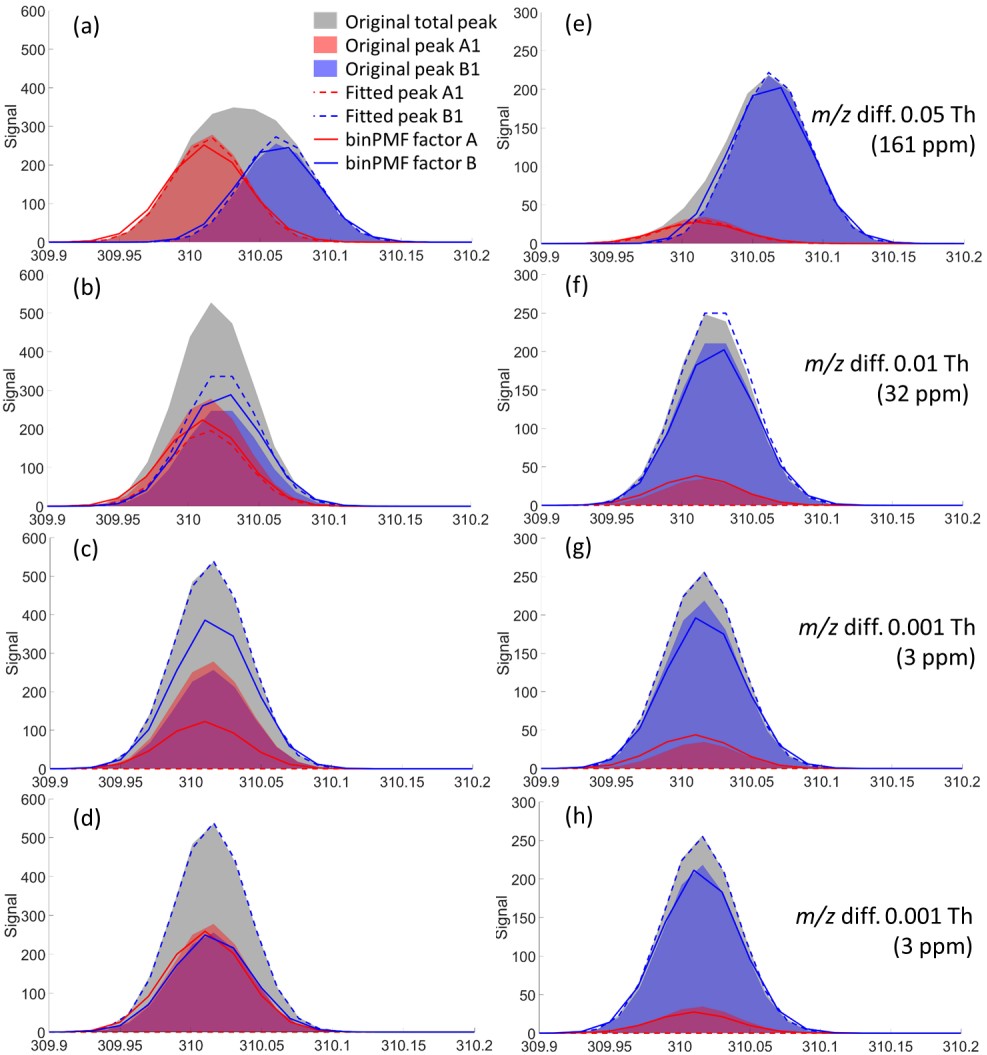

Figure 4. Peak separation results by a traditional HR fitting method (dashed lines) and binPMF (solid lines), at the 79[th] time point (panels a-d) and at the 21[st] time point (e-h) for experiment numbers 1 (a, e), 5 (b, f), 10 (c, g), and 20 (d, h). The signal intensity ratio of peaks A1 and B1 were about 1:1 and 1:6, respectively, at the 79[th] and the 21[st] time points. Panels a-c and e-g are for the one-mass system, while panels d and h are for the two-mass system.





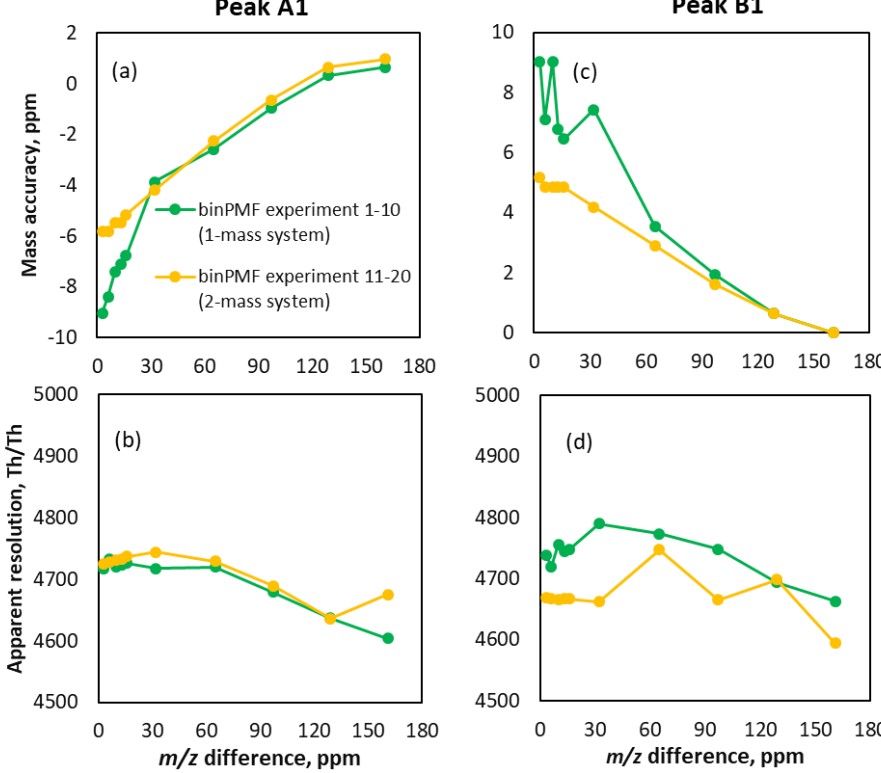


Figure 5. Characteristics of peaks fitted to binPMF factors. Panel a and b show results for peak A1,


and c and d for peak B1. In panels a and c, the mass accuracy of peaks resolved by binPMF are


compared to the original data. Panels b and d depict the resolution of the two fitted peaks. The original


resolution of the input data was 5000 Th/Th.




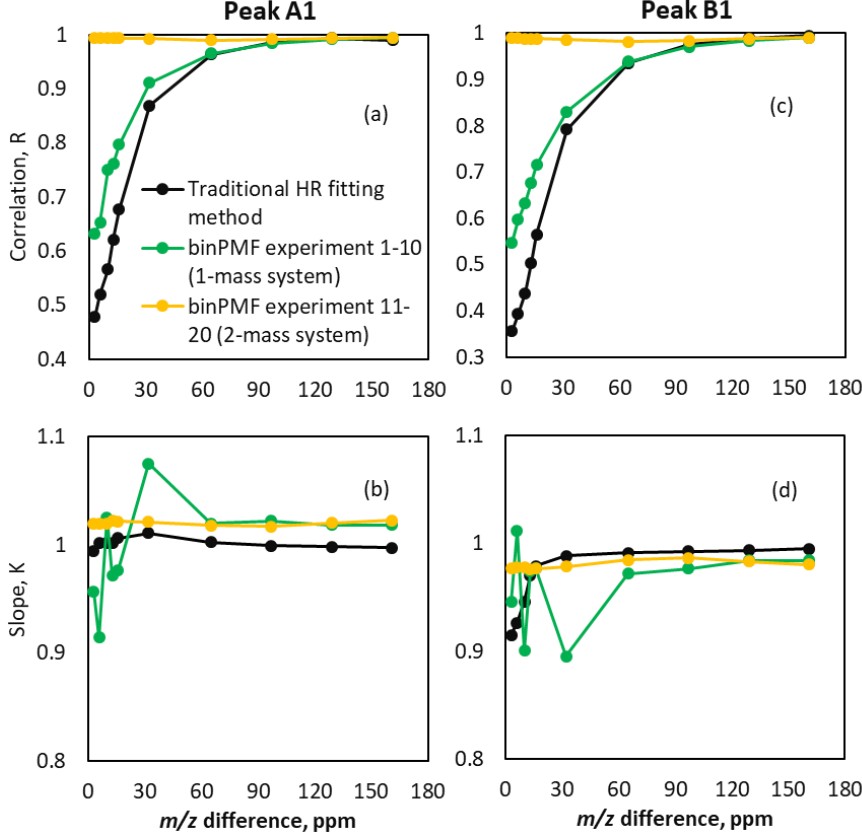

Figure 6. Comparison of time series of binPMF and HR fitting. Panel a and b show results for peak A1, and c and d for peak B1. Correlation of time series (panels a and c) retrieved by binPMF (green lines for experiments 1-10, yellow for 11-20) and traditional HR fitting (black lines) compared to original input data. Panels b and d depict the slope K of the linear fit y = k×x, where y is the signal retrieved from the synthetic data by either binPMF or the HR fitting, and x is the original input signals.




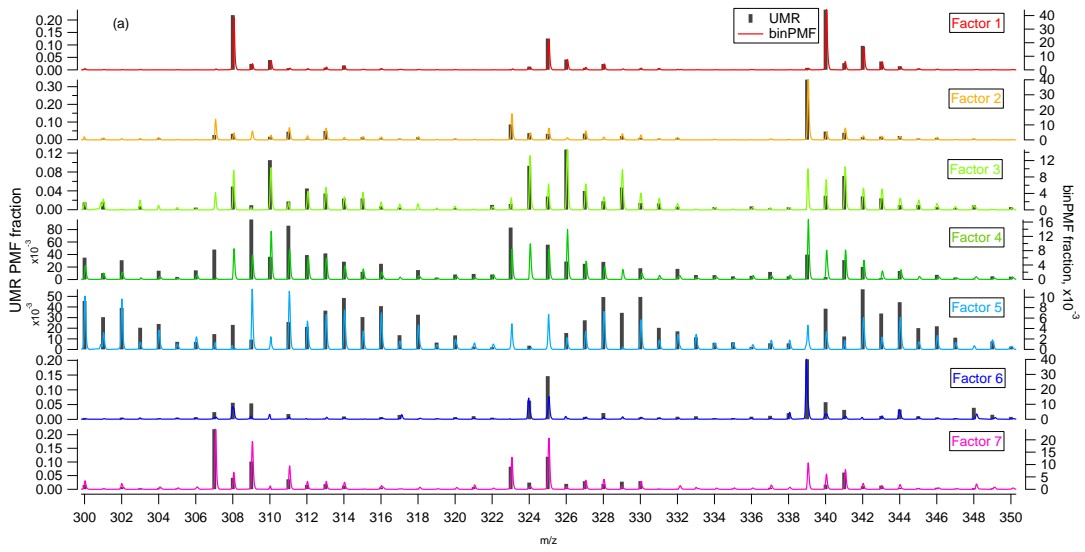


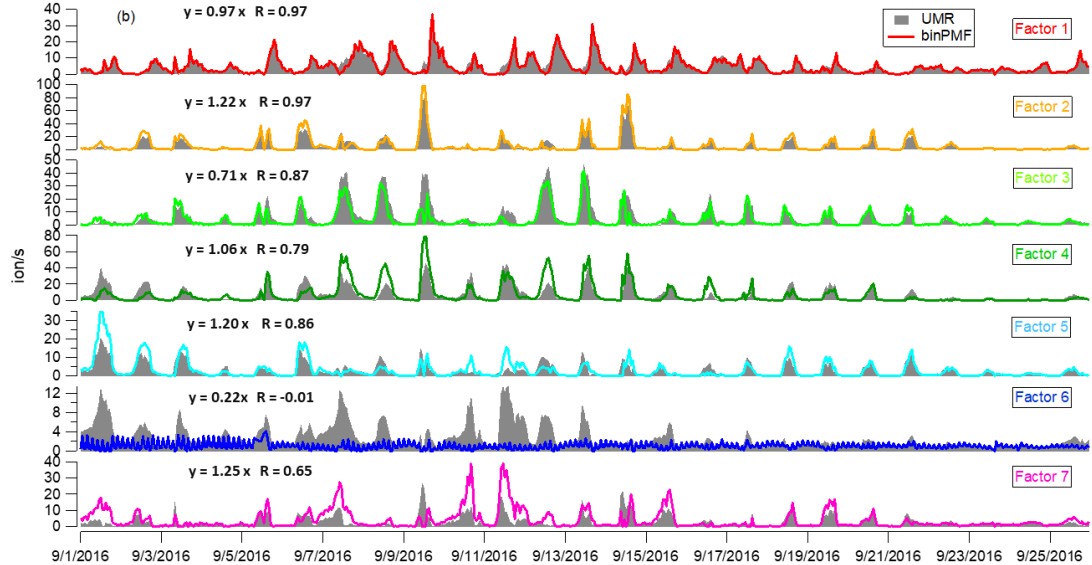

Figure 7. Comparison of binPMF and UMR-PMF for factor mass spectral profiles (panel a) and time
series (panel b). The equations in each panel describe how signals from binPMF (y) compare with
the UMR-PMF solution (x). R is the correlation coefficient between the time series.




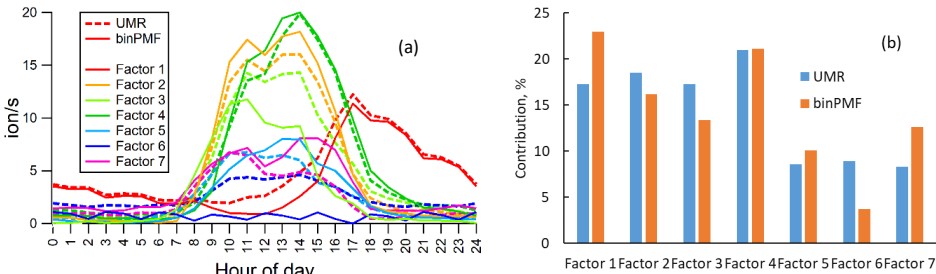


Figure 8. Comparison of binPMF and UMR-PMF for (a) diurnal trend and (b) factor contribution



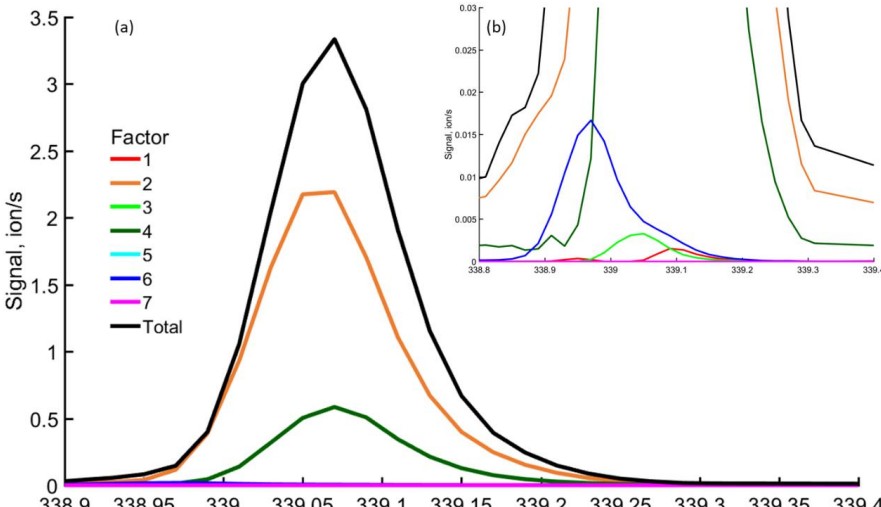


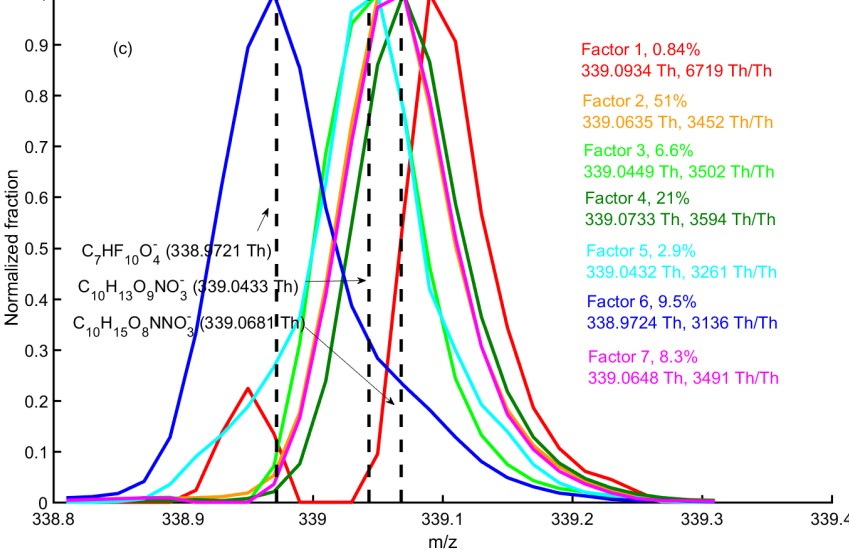


Figure 9. binPMF factor profiles at m/z 339 Th at 12:00 on September 9th. Panels a and b show the absolute concentrations of each factor, while in panel c, the factor profiles are normalized to the same maximum peak heights. The fitted peak location (Th) and the apparent resolution (Th/Th) for each factor is given in panel c,. The contribution of different factors to the integer *m/z* 339 Th is shown as a percentage. Three potential chemical compositions were marked with black vertical dashed lines.





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

A GLOBAL-MODEL OF NATURAL VOLATILE ORGANIC-COMPOUND EMISSIONS, Journal of
Geophysical Research-Atmospheres, 100, 8873-8892, 10.1029/94jd02950, 1995.
Hakola, H., Tarvainen, V., Bäck, J., Ranta, H., Bonn, B., Rinne, J., and Kulmala, M.: Seasonal variation of
mono- and sesquiterpene emission rates of Scots pine, Biogeosciences, 3, 93-101, 10.5194/bg-3-93-2006,
676 2006.
Hari, P., and Kulmala, M.: Station for Measuring Ecosystem–Atmosphere Relations (SMEAR II), Boreal
Environment Research, 10, 315-322, 2005.
Heinritzi, M., Simon, M., Steiner, G., Wagner, A. C., Kürten, A., Hansel, A., and Curtius, J.:
Characterization of the mass-dependent transmission efficiency of a CIMS, Atmos. Meas. Tech., 9, 1449-
1460, 10.5194/amt-9-1449-2016, 2016.
Henry, R. C.: Current factor analysis receptor models are ill-posed, Atmospheric Environment (1967), 21,
1815-1820, https://doi.org/10.1016/0004-6981(87)90122-3, 1987.



684 Huang, S., Rahn, K. A., and Arimoto, R.: Testing and optimizing two factor-analysis techniques on aerosol
685 at Narragansett, Rhode Island, Atmospheric Environment, 33, 2169-2185, https://doi.org/10.1016/S1352-
686 2310(98)00324-0, 1999.
687 Huey, L. G.: Measurement of trace atmospheric species by chemical ionization mass spectrometry:
688 Speciation of reactive nitrogen and future directions, Mass Spectrometry Reviews, 26, 166-184,
689 10.1002/mas.20118, 2007.
690 Jimenez, J. L., Canagaratna, M. R., Donahue, N. M., Prevot, A. S. H., Zhang, Q., Kroll, J. H., DeCarlo, P. F.,
691 Allan, J. D., Coe, H., Ng, N. L., Aiken, A. C., Docherty, K. S., Ulbrich, I. M., Grieshop, A. P., Robinson, A.
692 L., Duplissy, J., Smith, J. D., Wilson, K. R., Lanz, V. A., Hueglin, C., Sun, Y. L., Tian, J., Laaksonen, A.,
693 Raatikainen, T., Rautiainen, J., Vaattovaara, P., Ehn, M., Kulmala, M., Tomlinson, J. M., Collins, D. R.,
694 Cubison, M. J., Dunlea, J., Huffman, J. A., Onasch, T. B., Alfarra, M. R., Williams, P. I., Bower, K., Kondo,
695 Y., Schneider, J., Drewnick, F., Borrmann, S., Weimer, S., Demerjian, K., Salcedo, D., Cottrell, L., Griffin,
696 R., Takami, A., Miyoshi, T., Hatakeyama, S., Shimono, A., Sun, J. Y., Zhang, Y. M., Dzepina, K., Kimmel,
697 J. R., Sueper, D., Jayne, J. T., Herndon, S. C., Trimborn, A. M., Williams, L. R., Wood, E. C., Middlebrook,
698 A. M., Kolb, C. E., Baltensperger, U., and Worsnop, D. R.: Evolution of Organic Aerosols in the
699 Atmosphere, Science, 326, 1525-1529, 10.1126/science.1180353, 2009.
700 Jokinen, T., Sipilä, M., Junninen, H., Ehn, M., Lönn, G., Hakala, J., Petäjä, T., Mauldin Iii, R. L., Kulmala,
701 M., and Worsnop, D. R.: Atmospheric sulphuric acid and neutral cluster measurements using CI-APi-TOF,
702 Atmos. Chem. Phys., 12, 4117-4125, 10.5194/acp-12-4117-2012, 2012.
703 Jokinen, T., Berndt, T., Makkonen, R., Kerminen, V.-M., Junninen, H., Paasonen, P., Stratmann, F.,
704 Herrmann, H., Guenther, A. B., Worsnop, D. R., Kulmala, M., Ehn, M., and Sipila, M.: Production of
705 extremely low volatile organic compounds from biogenic emissions: Measured yields and atmospheric
706 implications, Proceedings of the National Academy of Sciences of the United States of America, 112, 7123-
707 7128, 10.1073/pnas.1423977112, 2015.
708 Junninen, H., Ehn, M., Petäjä, T., Luosujärvi, L., Kotiaho, T., Kostiainen, R., Rohner, U., Gonin, M., Fuhrer,
709 K., Kulmala, M., and Worsnop, D. R.: A high-resolution mass spectrometer to measure atmospheric ion
710 composition, Atmos. Meas. Tech., 3, 1039-1053, 10.5194/amt-3-1039-2010, 2010.
711 Kirkby, J., Duplissy, J., Sengupta, K., Frege, C., Gordon, H., Williamson, C., Heinritzi, M., Simon, M., Yan,
712 C., Almeida, J., Troestl, J., Nieminen, T., Ortega, I. K., Wagner, R., Adamov, A., Amorim, A., Bernhammer,
713 A.-K., Bianchi, F., Breitenlechner, M., Brilke, S., Chen, X., Craven, J., Dias, A., Ehrhart, S., Flagan, R. C.,
714 Franchin, A., Fuchs, C., Guida, R., Hakala, J., Hoyle, C. R., Jokinen, T., Junninen, H., Kangasluoma, J.,
715 Kim, J., Krapf, M., Kuerten, A., Laaksonen, A., Lehtipalo, K., Makhmutov, V., Mathot, S., Molteni, U.,
716 Onnela, A., Peraekylae, O., Piel, F., Petaejae, T., Praplan, A. P., Pringle, K., Rap, A., Richards, N. A. D.,
717 Riipinen, I., Rissanen, M. P., Rondo, L., Sarnela, N., Schobesberger, S., Scott, C. E., Seinfeld, J. H., Sipilae,
718 M., Steiner, G., Stozhkov, Y., Stratmann, F., Tome, A., Virtanen, A., Vogel, A. L., Wagner, A. C., Wagner,
719 P. E., Weingartner, E., Wimmer, D., Winkler, P. M., Ye, P., Zhang, X., Hansel, A., Dommen, J., Donahue,
720 N. M., Worsnop, D. R., Baltensperger, U., Kulmala, M., Carslaw, K. S., and Curtius, J.: Ion-induced
721 nucleation of pure biogenic particles, Nature, 533, 521-+, 10.1038/nature17953, 2016.
722 Kroll, J. H., Donahue, N. M., Jimenez, J. L., Kessler, S. H., Canagaratna, M. R., Wilson, K. R., Altieri, K. E.,
723 Mazzoleni, L. R., Wozniak, A. S., and Bluhm, H.: Carbon oxidation state as a metric for describing the
724 chemistry of atmospheric organic aerosol, Nature Chemistry, 3, 133, 2011.
725 Lanz, V. A., Alfarra, M. R., Baltensperger, U., Buchmann, B., Hueglin, C., Szidat, S., Wehrli, M. N.,
726 Wacker, L., Weimer, S., Caseiro, A., Puxbaum, H., and Prevot, A. S. H.: Source Attribution of Submicron
727 Organic Aerosols during Wintertime Inversions by Advanced Factor Analysis of Aerosol Mass Spectra,
728 Environmental Science & Technology, 42, 214-220, 10.1021/es0707207, 2008.
729 Lee, B. H., Lopez-Hilfiker, F. D., Mohr, C., Kurtén, T., Worsnop, D. R., and Thornton, J. A.: An Iodide-
730 Adduct High-Resolution Time-of-Flight Chemical-Ionization Mass Spectrometer: Application to
731 Atmospheric Inorganic and Organic Compounds, Environmental Science & Technology, 48, 6309-6317,
732 10.1021/es500362a, 2014.
733 Massoli, P., Stark, H., Canagaratna, M. R., Krechmer, J. E., Xu, L., Ng, N. L., Mauldin, R. L., Yan, C.,
734 Kimmel, J., Misztal, P. K., Jimenez, J. L., Jayne, J. T., and Worsnop, D. R.: Ambient Measurements of
735 Highly Oxidized Gas-Phase Molecules during the Southern Oxidant and Aerosol Study (SOAS) 2013, ACS
736 Earth and Space Chemistry, 10.1021/acsearthspacechem.8b00028, 2018.
737 Paatero, P., and Tapper, U.: Positive matrix factorization: A non-negative factor model with optimal
738 utilization of error estimates of data values, Environmetrics, 5, 111-126, 1994.



Paatero, P.: Least squares formulation of robust non-negative factor analysis, Chemometrics and Intelligent
Laboratory Systems, 37, 23-35, https://doi.org/10.1016/S0169-7439(96)00044-5, 1997.
Paatero, P.: The Multilinear Engine—A Table-Driven, Least Squares Program for Solving Multilinear
Problems, Including the n-Way Parallel Factor Analysis Model, Journal of Computational and Graphical
Statistics, 8, 854-888, 10.1080/10618600.1999.10474853, 1999.
Paatero, P., Hopke, P. K., Song, X.-H., and Ramadan, Z.: Understanding and controlling rotations in factor
analytic models, Chemometrics and Intelligent Laboratory Systems, 60, 253-264,
https://doi.org/10.1016/S0169-7439(01)00200-3, 2002.
Paatero, P., and Hopke, P. K.: Discarding or downweighting high-noise variables in factor analytic models,
Analytica Chimica Acta, 490, 277-289, https://doi.org/10.1016/S0003-2670(02)01643-4, 2003.
Polissar, A. V., Hopke, P. K., Paatero, P., Malm, W. C., and Sisler, J. F.: Atmospheric aerosol over Alaska:
2. Elemental composition and sources, Journal of Geophysical Research: Atmospheres, 103, 19045-19057,
751  1998.
Pope III, C. A., Ezzati, M., and Dockery, D. W.: Fine-particulate air pollution and life expectancy in the
United States, New England Journal of Medicine, 360, 376-386, 2009.
Schauer, J. J., Rogge, W. F., Hildemann, L. M., Mazurek, M. A., Cass, G. R., and Simoneit, B. R.: Source
apportionment of airborne particulate matter using organic compounds as tracers, Atmospheric Environment,
756  30, 3837-3855, 1996.
Shiraiwa, M., Ueda, K., Pozzer, A., Lammel, G., Kampf, C. J., Fushimi, A., Enami, S., Arangio, A. M.,
Fröhlich-Nowoisky, J., Fujitani, Y., Furuyama, A., Lakey, P. S. J., Lelieveld, J., Lucas, K., Morino, Y.,
Pöschl, U., Takahama, S., Takami, A., Tong, H., Weber, B., Yoshino, A., and Sato, K.: Aerosol Health
Effects from Molecular to Global Scales, Environmental Science & Technology, 51, 13545-13567,
10.1021/acs.est.7b04417, 2017.
Song, Y., Shao, M., Liu, Y., Lu, S., Kuster, W., Goldan, P., and Xie, S.: Source apportionment of ambient
volatile organic compounds in Beijing, Environmental science & technology, 41, 4348-4353, 2007.
Stark, H., Yatavelli, R. L. N., Thompson, S. L., Kimmel, J. R., Cubison, M. J., Chhabra, P. S., Canagaratna,
M. R., Jayne, J. T., Worsnop, D. R., and Jimenez, J. L.: Methods to extract molecular and bulk chemical
information from series of complex mass spectra with limited mass resolution, International Journal of Mass
Spectrometry, 389, 26-38, https://doi.org/10.1016/j.ijms.2015.08.011, 2015.
Stocker, T., Qin, D., Plattner, G., Tignor, M., Allen, S., Boschung, J., Nauels, A., Xia, Y., Bex, V., and
Midgley, P.: IPCC, 2013: Climate Change 2013: The Physical Science Basis. Contribution of Working
Group I to the Fifth Assessment Report of the Intergovernmental Panel on Climate Change, 1535 pp, in,
Cambridge Univ. Press, Cambridge, UK, and New York, 2013.
Sun, Y.-L., Zhang, Q., Schwab, J., Demerjian, K., Chen, W.-N., Bae, M.-S., Hung, H.-M., Hogrefe, O.,
Frank, B., and Rattigan, O.: Characterization of the sources and processes of organic and inorganic aerosols
in New York city with a high-resolution time-of-flight aerosol mass apectrometer, Atmospheric Chemistry
and Physics, 11, 1581-1602, 2011.
Ulbrich, I. M., Canagaratna, M. R., Zhang, Q., Worsnop, D. R., and Jimenez, J. L.: Interpretation of organic
components from Positive Matrix Factorization of aerosol mass spectrometric data, Atmos. Chem. Phys., 9,
2891-2918, 10.5194/acp-9-2891-2009, 2009.
Wei, W., Wang, S., Chatani, S., Klimont, Z., Cofala, J., and Hao, J.: Emission and speciation of non-methane
volatile organic compounds from anthropogenic sources in China, Atmospheric Environment, 42, 4976-
4988, https://doi.org/10.1016/j.atmosenv.2008.02.044, 2008.
Yan, C., Nie, W., Aijala, M., Rissanen, M. P., Canagaratna, M. R., Massoli, P., Junninen, H., Jokinen, T.,
Sarnela, N., Hame, S. A. K., Schobesberger, S., Canonaco, F., Yao, L., Prevot, A. S. H., Petaja, T., Kulmala,
M., Sipila, M., Worsnop, D. R., and Ehn, M.: Source characterization of highly oxidized multifunctional
compounds in a boreal forest environment using positive matrix factorization, Atmospheric Chemistry and
Physics, 16, 12715-12731, 10.5194/acp-16-12715-2016, 2016.
Zha, Q., Yan, C., Junninen, H., Riva, M., Sarnela, N., Aalto, J., Quéléver, L., Schallhart, S., Dada, L.,
Heikkinen, L., Peräkylä, O., Zou, J., Rose, C., Wang, Y., Mammarella, I., Katul, G., Vesala, T., Worsnop, D.
R., Kulmala, M., Petäjä, T., Bianchi, F., and Ehn, M.: Vertical characterization of highly oxygenated
molecules (HOMs) below and above a boreal forest canopy, Atmos. Chem. Phys., 18, 17437-17450,
10.5194/acp-18-17437-2018, 2018.





Zhang, Q., Jimenez, J. L., Canagaratna, M., Allan, J., Coe, H., Ulbrich, I., Alfarra, M., Takami, A.,
Middlebrook, A., and Sun, Y.: Ubiquity and dominance of oxygenated species in organic aerosols in
anthropogenically-influenced Northern Hemisphere midlatitudes, Geophysical Research Letters, 34, 2007.
Zhang, Q., Jimenez, J. L., Canagaratna, M. R., Ulbrich, I. M., Ng, N. L., Worsnop, D. R., and Sun, Y.:
Understanding atmospheric organic aerosols via factor analysis of aerosol mass spectrometry: a review,
Analytical and Bioanalytical Chemistry, 401, 3045-3067, 10.1007/s00216-011-5355-y, 2011.
Zhang, Y., Lin, Y., Cai, J., Liu, Y., Hong, L., Qin, M., Zhao, Y., Ma, J., Wang, X., and Zhu, T.: Atmospheric
PAHs in North China: spatial distribution and sources, Science of the Total Environment, 565, 994-1000,
800    2016.
Zhang, Y., Cai, J., Wang, S., He, K., and Zheng, M.: Review of receptor-based source apportionment
research of fine particulate matter and its challenges in China, Science of the Total Environment, 586, 917-
803    929, 2017.
