# Peer review of "A Novel Approach for Simple Statistical Analysis of High-Resolution"

_Atmospheric Measurement Techniques, 2019_

## Referee Comment (RC1) · Anonymous Referee #2 · 17 Apr 2019

This paper introduces a new method of interpreting ambient mass spectrometer data, binPMF. This is an extension of existing PMF techniques to sub-unity resolution mass spectra without the usual peak fitting stage before this. The advantage of this approach is that it can utilise the extra mass spectrometric information without having to specify the peaks that are expected to be present. I can foresee a number of applications for this technique; it could conceivably be used deliver more accurate factorisations than UMR-PMF, but I strongly suspect that it may prove more useful in identifying peaks to use for HR-PMF. However, time will tell on that.

Presented as a proof-of-concept, this is likely to stimulate activity and further development within the community of users of ARI/Tofwerk instruments, but I would anticipate that this may have applications beyond this. The paper is certainly relevant to AMT and

is very well written and presented. I have a few reservations (see below) but these are minor, so subject to these I recommend publication.

Comments:

Generally, the authors are very bullish about the capabilities of the technique, but I can foresee a number of fundamental limitations. In the interests of properly exploring this method on a conceptual level, I would recommend that the authors discuss these so as to create realistic expectations of what this is capable of. The issues I can think of are as follows, but there may be others:

* Peak shape - It must be explicitly stated that this technique assumes that the peak shape model is consistent throughout a dataset. While this should remain constant if the instrument is working properly, if it drifts somehow, then this will likely cause unpredictable behaviour here.

* Aliasing - the act of binning the mass spectra through over- followed by under-sampling may introduce artificial smoothing of the data. This is unlikely to be an issue for peaks at low m/z ratios where the fundamental resolution of the instrument is high, but where the resolution of the mass spectrometer starts to become comparable to the target bin width, I imagine this could be an issue. This would be unlikely to cause problems if the mass calibration of the spectrometer were constant, but as this is known to subtly drift over time, this means that any aliasing artefacts could (in theory) be variable with time, even if the changes in calibration were properly accounted for, which in turn could create artificial factors in the dataset. This should be discussed, if only conceptually.

* Complexity - the analysis may not be able to adequately factorise systems where there is a large number of degrees of freedom in the chemistry, e.g. studying SOA formation in a chamber. This is a fundamental limitation of PMF and applies equally to the other established methods as well. However, I suppose this technique could still be of use in peak identification, even if it can't explain all the variance.

* Covariance - as with all PMF, the ability to separate components is contingent on them showing different trends. If (hypothetically) two adjacent peaks were to covary, then this technique would fail to separate them.

Line 196: If the noise is based on signal-free regions of the mass spectrum, would this not be underestimated because of the thresholding applied by the data acquisition system?

Line 535: I disagree that mass calibration could be accounted for by an error term; according to the PMF data model, the errors are supposed to be random and independent of one another, however a shift in mass calibration would cause deviations that are dependent on adjacent points.
* * *

---

## Referee Comment (RC2) · Anonymous Referee #1 · 18 May 2019

This manuscript reports a novel "binPMF" method that can be used to improve the deconvolution of organic factors using atmospheric mass spectral data that don't have adequate mass resolution for unambiguous ion speciation. The authors applied this method to both ambient and synthetic data and demonstrated that the combination of mass spectral binning with positive matrix factorization is an effective approach to better resolve chemical information and to improve the separation of different sources and processes. This is an exciting new development in data analysis for real-time mass spectrometry and this work is of very high quality. The manuscript is very well written and the topic is a good fit for AMT. I thus recommend the manuscript be accepted for publication as is.

---

## Author Comment (AC1) · 11 Jun 2019

**Responses to reviewers' comments**

We would like to thank the reviewers for their valuable and constructive feedbacks and comments, which helped us to improve the manuscript.

Referee comments are given in black, and our responses are in blue. Changes made to the manuscript are marked in underlined blue. The line number referred here is for the new revised manuscript.

**Anonymous Referee #1**

This manuscript reports a novel "binPMF" method that can be used to improve the deconvolution of organic factors using atmospheric mass spectral data that don't have adequate mass resolution for unambiguous ion speciation. The authors applied this method to both ambient and synthetic data and demonstrated that the combination of mass spectral binning with positive matrix factorization is an effective approach to better resolve chemical information and to improve the separation of different sources and processes. This is an exciting new development in data analysis for real-time mass spectrometry and this work is of very high quality. The manuscript is very well written and the topic is a good fit for AMT. I thus recommend the manuscript be accepted for publication as is.

**Response:** We greatly appreciate this positive feedback on our work.

---

## Author Comment (AC2) · 11 Jun 2019

**Responses to reviewers' comments**

We would like to thank the reviewers for their valuable and constructive feedbacks and comments, which helped us to improve the manuscript.

Referee comments are given in black, and our responses are in blue. Changes made to the manuscript are marked in underlined blue. The line number referred here is for the new revised manuscript.

**Anonymous Referee #2**

This paper introduces a new method of interpreting ambient mass spectrometer data, binPMF. This is an extension of existing PMF techniques to sub-unity resolution mass spectra without the usual peak fitting stage before this. The advantage of this approach is that it can utilise the extra mass spectrometric information without having to specify the peaks that are expected to be present. I can foresee a number of applications for this technique; it could conceivably be used deliver more accurate factorisations than UMR-PMF, but I strongly suspect that it may prove more useful in identifying peaks to use for HR-PMF. However, time will tell on that. Presented as a proof-of-concept, this is likely to stimulate activity and further development within the community of users of ARI/Tofwerk instruments, but I would anticipate that this may have applications beyond this. The paper is certainly relevant to AMT and is very well written and presented. I have a few reservations (see below) but these are minor, so subject to these I recommend publication.

Comments: Generally, the authors are very bullish about the capabilities of the technique, but I can foresee a number of fundamental limitations. In the interests of properly exploring this method on a conceptual level, I would recommend that the authors discuss these so as to create realistic expectations of what this is capable of. The issues I can think of are as follows, but there may be others:

**Response:** We thank the reviewer for the comments and will answer them point-by-point below. We acknowledge already at this stage that our aim was not to present binPMF as a method without limitations. For example, any fundamental limitations inherent to PMF will obviously still apply to binPMF. Instead, we want here to emphasize the word "simple" in our title. HR fitting of spectra can be extremely laborious, and with enough ions at a single integer mass, HR fitting can become impossible. BinPMF can still be run in such cases, with data preparation requiring the same amount of work as for a simpler data set, informing the analyst about various aspects of his/her data, while still utilizing more spectral information than simply running UMR PMF.

\* Peak shape - It must be explicitly stated that this technique assumes that the peak shape model is consistent throughout a dataset. While this should remain constant if the instrument is working properly, if it drifts somehow, then this will likely cause unpredictable behaviour here.

**Response:** For the technique of binPMF itself, it only tries to separate different bins based on their temporal behaviors, and the narrow bins help to preserve the high-resolution information. Thus, the technique does not assume anything about the peak shape. However, the reviewer is correct that in the way we present the results and their interpretation, we as analysts have assumed a constant peak shape throughout our data sets. Any variability in the peak shape, or resolution for that matter, will cause signal to "leak" different amounts into adjacent bins at different times. The larger this variation is, the harder the factor and peak separation becomes in binPMF.

An inconsistent peak shape in a data set will be a much larger issue for traditional peak fitting, and an invariable peak shape is a basic assumption in the routines of tofTools or Tofware. With problematic peak shape issues, only UMR results will be completely unaffected, but binPMF will be less affected, and thus likely provide a better result, compared to any method based on high-resolution peak fitting.

Ultimately, variable peak shapes over time are an indication that some large change has occurred in the instrument, and data before and after this change should likely not be analyzed together in the first place. Luckily, in our experience, the peak shape typically stays very consistent.

We added the following sentence (underlined) on Line 545-547 in Section 3.3 in the manuscript in order to briefly discuss this issue about peak shape:

"…… variation could also be considered for error estimation. Similarly, although generally rare and suggestive of some instrumental problem, if the peak shape or resolution shift over time, this would also require an improved error estimation in order to account for increased variability."

* Aliasing - the act of binning the mass spectra through over- followed by under-sampling may introduce artificial smoothing of the data. This is unlikely to be an issue for peaks at low m/z ratios where the fundamental resolution of the instrument is high, but where the resolution of the mass spectrometer starts to become comparable to the target bin width, I imagine this could be an issue. This would be unlikely to cause problems if the mass calibration of the spectrometer were constant, but as this is known to subtly drift over time, this means that any aliasing artefacts could (in theory) be variable with time, even if the changes in calibration were properly accounted for, which in turn could create artificial factors in the dataset. This should be discussed, if only conceptually.

Response: In the binning process of binPMF, the data are first linearly interpolated with certain interpolation interval, and then binned with certain bin width. The bin width is much larger than the interpolation interval. The binning process considers all the interpolated data points within the bin width by averaging their signals, instead of just taking one data point as representative for this bin. In addition, in the manuscript section 3.3., we also suggest that the bin width should be mass-dependent and defined in such way that each peak can be covered by 7 bins, considering the instrument resolution (Th/Th).

While any resampling of data will inevitably lead to some smoothing, it should be minor with this approach. For the aliasing effect suggested by the reviewer, the fact that the bin widths are always clearly narrower than the peak widths, should make this effect negligible. We added a note on this into Section 3.3 on Line 516:

"Too few bins per peak would mean that we may lose valuable information in the binning, and potentially risk introducing aliasing effects, while too many points……"

* Complexity - the analysis may not be able to adequately factorise systems where there is a large number of degrees of freedom in the chemistry, e.g. studying SOA formation in a chamber. This is a fundamental limitation of PMF and applies equally to the other established methods as well. However, I suppose this technique could still be of use in peak identification, even if it can't explain all the variance.

Response: Yes, we agreed with the reviewer that factorization analysis cannot separate every pathway/source. PMF itself, as a factorization analysis method, has its own fundamental limitations and problems and binPMF cannot improve on these problems. We added one sentence (underlined) in the manuscript in Line 173-174 in Section 2.1,

"……considering their rotational uniqueness. Finally, we note that in addition to rotational ambiguity, binPMF also inherits all other fundamental limitations and strengths of the underlying PMF method."

* Covariance - as with all PMF, the ability to separate components is contingent on them showing different trends. If (hypothetically) two adjacent peaks were to covary, then this technique would fail to separate them.

**Response:** Yes, the reviewer is correct. However, we do not find this to be a problem, and it would not impact the separation of factors.

Even so, for example, if there are $C_{10}H_{16}O_7$ and $C_9H_{12}O_8$ in the same mass spectrum, which will go to the same factor, we can still get the HR information in this factor, as it will now be a double peak at m/z of 310 Th in the factor. Thus, compared to UMR, we get more information. Compared to traditional HR fitting, assuming one knows the two ions are present at this mass, they can be fitted and separated, but will not contribute much more to the separation of different factors than the binPMF approach. In addition, if the two adjacent peaks are very close in mass, the attribution of signal between the two fitted peaks may become variable. This, in turn, may cause covarying ions to have different, anti-correlating behavior from improper HR fitting. This, again, highlights the robustness of binPMF, which requires no identification of ions beforehand, and is much less sensitive to variations in mass calibration, resolution or peak shapes.

Line 196: If the noise is based on signal-free regions of the mass spectrum, would this not be underestimated because of the thresholding applied by the data acquisition system?

**Response:** For the error estimation, we use this equation $S_{ij} = \sigma_{ij} + \sigma_{noise}$ .The signal dependent part $\sigma_{ij}$ represents the poisson counting error. The noise term $\sigma_{noise}$ is to represent the background noise, stray counts etc. Exactly how good our method of noise estimation is, will depend on a wide variety of factors, including the frequency of ions measured at a given mass, the type of data acquisition card (compare e.g. TDC vs ADC), the ratio of the threshold level compared to the electronic noise, etc. If the reviewer is suggesting that the electronic noise is "thresholded away" in the signal-free region, but that this noise is superimposed on the signal when an actual ion is detected, then there are a few effects counteracting this potential problem. At low count rates a given bin, most acquired spectra (sampled at ~10 kHz) will lack ions, and the majority of noise will arise from stray counts in the spectra, which were signal-free in reality. When the count rate becomes higher, the spectra with real signal become more dominant, but at this stage the signal-dependent error term $\sigma_{ij}$ will start to dominate the total error. In other words, for low (or no) signal bins, and for high signal bins, the error estimation approach is expected to work correctly. For intermediate count rates, there might be an effect as suggested by the reviewer. However, this cannot be addressed by simply increasing the noise term, since this would cause the low signals to have over-estimated noise. Thus, this simple error model is concluded to be adequate for the purpose of binPMF.

Line 535: I disagree that mass calibration could be accounted for by an error term; according to the PMF data model, the errors are supposed to be random and independent of one another, however a shift in mass calibration would cause deviations that are dependent on adjacent points.

**Response:** Uncertainty in PMF analyses arises from three main causes, as described in Paatero et al., 2014: (1) random errors in data values; (2) rotational ambiguity; and (3) modeling errors. This question by the reviewer is discussed in great detail in Paatero et al., 2014. Variations in mass calibration are one example

of a modeling error. The assumptions underlying PMF are completely fulfilled if there are no modelling errors, and this is often the case in simulation studies. Then the results obtained from PMF obey rather strict mathematical rules, e.g. the Q contributions from individual columns or individual rows obey rather narrow chi-square distributions.

However, in real data, modeling errors are usually present, often in significant amounts. This means that statistical properties of PMF results are undefined. On the other hand, PMF analysis may still be used even in presence of modeling errors. It is common practice to include expected effect of modeling errors of data values so that the uncertainties $S_{ij}$ specified for erroneous data values are artificially increased. In this way, the effect of modeling errors is constrained although it cannot be totally eliminated.

We added the following sentences in Line 536-544 in Section 3.3 in the manuscript,

"Uncertainty in PMF analyses arises from three main causes, random errors in data values, rotational ambiguity, and modeling errors (Paatero et al., 2014). Variations in mass calibration are one example of a modeling error, It is common practice to increase uncertainty values $S_{ij}$ specified for data values disturbed by modeling errors. This increase does not account for the mass calibration error in the sense that the effect of mass calibration variation would disappear. The increase simply balances residuals in different data values so that the best possible result may be obtained. In addition to the two error estimation terms discussed in section 2.3, $\sigma_{ij}$ and $\sigma_{noise}$, a third form of error, to balance the mass calibration variation could also be considered for error estimation."

Reference

Paatero, P., Eberly, S., Brown, S. G., and Norris, G. A.: Methods for estimating uncertainty in factor analytic solutions, Atmos. Meas. Tech., 7, 781-797, doi:10.5194/amt-7-781-2014, 2014.